# The multi-drug resistant organisms infections decrease during the antimicrobial stewardship era in cirrhotic patients: An Italian cohort study

Andrea Dalbeni[1,2], Anna Mantovani[1,2]*, Mirko Zoncapè[1,2], Filippo Cattazzo[1,2], Michele Bevilacqua[1,2], Leonardo De Marco[1,2], Veronica Paon[2], Donatella Ieluzzi[2], Anna Maria Azzini[3], Elena Carrara[3], Evelina Tacconelli[3], David Sacerdoti[2]

1 Division of General Medicine C, Department of Medicine, University and Azienda Ospedaliera Universitaria Integrata of Verona, Verona, Italy, 2 Liver Unit, Department of Medicine, University and Azienda Ospedaliera Universitaria Integrata of Verona, Verona, Italy, 3 Division of Infectious Disease, Department of Diagnostics and Public Health, University and Azienda Ospedaliera Universitaria Integrata of Verona, Verona, Italy

* anna.mantovani@aovr.veneto.it

**Data Availability Statement:** All relevant data are within the manuscript and its Supporting information files. Because of privacy issues all the other data are available upon request to our Ethics

## Abstract

### Background and purpose

Bacterial infections represent a major cause of morbidity and mortality in cirrhotic patients. Our aim was to assess the incidence of bacterial infections, in particular due to multidrug-resistant organisms (MDROs) before and after the introduction of the antimicrobial steward-ship program, "Stewardship Antimicrobial in VErona" (SAVE). In addition, we also analysed the liver complications and the crude mortality during the whole follow up.

### Methods

We analysed 229 cirrhotic subjects without previous hospitalization for infections enrolled at the University Verona Hospital from 2017 to 2019 and followed up until December 2021 (mean follow-up 42.7 months).

### Results

101 infections were recorded and 31.7% were recurrent. The most frequent were sepsis (24.7%), pneumonia (19.8%), spontaneous bacterial peritonitis (17.8%). 14.9% of infections were sustained by MDROs. Liver complications occurred more frequently in infected patients, and in case of MDROs infections with a significantly higher MELD and Child-Pugh score. In Cox regression analysis, mortality was associated with age, diabetes and bacterial infections episodes (OR 3.30, CI 95%: (1.63–6.70). Despite an increase in total infections over the past three years, a decrease in the incidence rate in MDROs infections was documented concurrently with the introduction of SAVE (IRD 28.6; 95% CI: 4.6–52.5, p = 0.02).

Committee (CESC VR-RO) for researchers who meet the criteria for access to confidential data. - e-mail contact supporto.noprofit@aovr.veneto.it - telephone contact +390458127043-7051).

**Funding:** The authors received no specific funding for this work.

**Competing interests:** The authors have declared that no competing interests exist.

**Abbreviations:** ACLF, acute-on-chronic liver failure; AIDS, acquired immune deficiency syndrome; AS, antimicrobial stewardship; CFU, colony-forming unit; CLD, chronic liver disease; COPD, chronic obstructive pulmonary disease; CP, Child-Pugh; CRE, carbapenem-resistant Enterobacterales; CRP, c-reactive protein; CT, computed tomography; CVC, central venous catheters; ESBL, extended spectrum beta lactamase; HCC, hepatocellular carcinoma; HE, hepatic encephalopathy; HIV, human immunodeficiency virus; HRS, hepatorenal syndrome; IQR, interquartile range; IRD, incidence rate difference; MDROs, multi-drug resistant organisms; MELD, model for end-stage liver disease; MR, methicillin-resistant; MRI, magnetic resonance imaging; MRSA, methicillin-resistant *Staphylococcus aureus*; PCT, procalcitonin; PMN, polymorphonuclear; PVT, portal vein thrombosis; SAVE, stewardship antimicrobial in Verona; SBP, spontaneous bacterial peritonitis; SD, standard deviation; SOFA, sequential organ failure assessment; SSTI, skin and soft tissues infections; US, ultrasonography; UTI, urinary tract infections; VRE, vancomycin-resistant Enterococci.

## Conclusions

Our study confirms the burden of bacterial infections in cirrhotic patients, especially MDROs, and the strong interconnection with liver complications. The introduction of SAVE decreased MDROs infections. Cirrhotic patients require a closer clinical surveillance to identify colonized patients and avoid the horizontal spread of MDROs in this setting.

## Introduction

Liver cirrhosis represents the final stage of chronic liver diseases (CLD) in which a compensated and a decompensated phase are distinguished [1, 2]. The transition between these two phases occurs overall with an incidence of 5–7% per year, reducing the survival from a median of 12 years to 2 years [1]. The development of decompensated cirrhosis is influenced by the persistence of etiological factors, recurrent gastrointestinal bleeding, renal impairment and especially infections [3–8], accounting for a 4-fold increase in mortality [9–12].

Cirrhotic patients are considered at increased risk to develop bacterial infections, because of their typical immune dysfunction syndrome and a more frequent bacterial exposure due to enteric bacterial translocation due to portal hypertension, gut dysbiosis [13] and a higher exposure to antibiotics, repeated invasive procedures and multiple hospitalizations [14]. The overall prevalence of bacterial infections in decompensated cirrhotic patients has been estimated between 30 and 46% [15]. Some studies have shown that 32–50% are community acquired [16, 17], while 25–41% are healthcare associated and 25–37% are related to hospitalization [18].

Moreover, almost 25% of cirrhotic patients with a bacterial infection are prone to develop a second infection, with a serious worsening of the clinical condition and, subsequently, an increase in the mortality risk, in particular in case of infections sustained by multi-drug resistant organisms (MDROs) [19].

The most frequent bacterial infections in cirrhotic patients are spontaneous bacterial peritonitis (SBP; 20–30%), urinary tract infections (UTI; 20–25%), pneumonia (8–20%), sepsis (8–20%) and skin and soft tissues infections (SSTI; 5–10%) [16–19]. SBP and UTI are sustained in 50–70% of cases by Gram negative bacteria, mainly Enterobacterales (*E. coli* and *K. pneumoniae*), while Gram positive bacteria are more often responsible for pneumonia and SSTI [17].

In the last decades, antimicrobial resistance has been recognized as one of the leading causes of death around the world. Globally, the prevalence of MDROs has been estimated around 34%, with significant differences between different geographic areas and even within the same region [17].

Inadequate empiric antibiotic therapy is the main predictor of short-term mortality in cirrhotic patients with a bacterial infection [20], therefore the selection of an adequate empirical antibiotic therapy is crucial to improve survival. However, a careful balance between adequate coverage and risk of antimicrobial resistance selection and toxicities should always guide antibiotic choices, in accordance with antimicrobial stewardship principles. These principles are mainly based on prevention of antibiotic misuse/overuse, with established duration of therapy, early de-escalation when indicated, and especially periodic review of local epidemiology [18].

The primary aim of the study was to assess in our cohort of cirrhotic patients the local infectious epidemiology, incidence and risk of bacterial infections (both community and healthcare associated) during a 5-years study period. Secondary aims included risk of infection due to MDROs, the association with liver complications, crude mortality and report of the effects of a local stewardship program (called "Stewardship Antimicrobial in VErona"- SAVE) on infections, focusing on MDROs.

## Patients and methods

This is a single-centre cohort study, approved by the local Institutional Ethics Committee (2730CESC-VR), in accordance with the Ethical Principles for Medical Research Involving Human Subjects outlined in the 2013 Declaration of Helsinki. We enrolled 229 cirrhotic patients without previous hospitalization for infections, referred to the Liver Unit of the Verona University Hospital (outpatients or inpatients), between January 2017 and 2019, and followed until December 2021, for a mean follow-up time of 42.7 ± 17.1 months. All patients signed a written consent at recruitment.

Inclusion criteria were: age ≥18 years, liver cirrhosis diagnosis assessed by ultrasonography (US), computed tomography (CT) scan or magnetic resonance imaging (MRI) and/or histologically; no previous hospitalization for infection; signing of informed consent form. Patients affected by clinical conditions predisposing to immunosuppression (HIV positive or AIDS confirmed; affected by primary immunodeficiency; solid organ recipients or patients with immunosuppressive therapies), hematological diseases, patients with previous severe infections and patients with cancer except for hepatocellular carcinoma (HCC), were excluded. We also excluded patients admitted for SARS-CoV-2 infection.

## Definition of bacterial infection

Infections were defined as follow: 1) SBP, presence of a polymorphonuclear (PMN) cell count of ≥250 cells/mm$^3$ on ascitic fluid; 2) secondary bacterial peritonitis, PMN cell count of ≥250 cells/ mm$^3$ in the ascitic fluid and radiological or surgical evidence of an infectious source in the abdomen; 3) bacteraemia with sepsis, blood cultures positive for clinically relevant bacteria in a patient with a sequential organ failure assessment (SOFA) score ≥ 2 at the time of diagnosis (in case of *S. haemolyticus* and *S. epidermidis*, when isolated in at least 2 consecutive sets of blood cultures); 4) UTI, presence of >10 leukocytes/field in urinary analysis and/or a positive urine culture associated with the presence of urinary symptoms (dysuria, stranguria, urgency, pollakiuria or a general increased urination frequency); 5) respiratory tract infections, clinical signs of lower respiratory tract infection with evidence of a new onset consolidation on chest X-ray and/or positive sputum's culture; 6) acute infectious cholangitis, increased cholestasis, pain in the right hypochondrium, jaundice and/or fever, with/ without evidence of biliary obstruction on radiological imaging; 7) SSTI, clinical signs of infection in association with signs of skin inflammation; 8) any suspected bacterial infection, presence of fever (measured forehead, axillary or tympanic temperature ≥37.5°C) and leukocytosis (>12,000 leukocytes/mm$^3$) and abnormal levels of CRP/PCT without identification of the infectious source; 9) bacterial gastroenteritis, defined by acute diarrhoea with a positive fecal culture.

Other conditions, not considered among infections, were defined as follows: 1) asymptomatic bacteriuria, a positive urine culture with bacterial count ≥10$^5$ CFU/mL in the absence of pyuria and/or urinary symptoms suggestive for infection; 2) enteric colonization by multidrug resistant bacteria, detected performing a surveillance rectal swab on hospital admission; 3) bacterascites, bacteria detected in ascitic fluid with a non-significant PMN cell count on ascitic fluid (< 250 cells/ mm$^3$).

Infections were considered community acquired if diagnosed at the hospital admission or developed within 48 hours and in absence of a history of hospitalization within the previous three months. Infections were considered nosocomial acquired if clinically documented after 48 hours of hospitalization [21], and healthcare-associated infections (HAIs), if acquired within 90 days prior to the hospital admission, during the hospitalization in another hospital or in another healthcare setting [22].

Microbiological exams were performed during hospitalization, if an ongoing bacterial infection was clinically suspected, in asymptomatic patients as surveillance of MDROs and during the work-up for liver transplantation. The microbiological tests considered were ascitic fluid cultures (performed bedside with the inoculation of at least 10 ml of ascitic fluid into blood culture bottles in order to increase their sensitivity to identify the microbiological agent), and blood cultures, both performed before starting an antibiotic therapy. We also collected urine cultures, stool cultures; moreover, rectal swabs were performed at admission to correctly identify vancomycin-resistant Enterococci (VRE) and carbapenem-resistant Enterobacterales (CRE) colonized patients. All patients resulted positive to a rectal swab for MDROs were kept isolated or cohorted.

Isolated micro-organisms were defined as MDR gram negative bacteria if resistant to at least one drug in at least three different classes of antibiotics, while for gram positive bacteria oxacillin-resistance or vancomycin-resistance were considered for Staphylococci or Enterococci, respectively.

## Antimicrobial Stewardship Verona

SAVE, "Stewardship Antibiotic VErona", was started at the Verona University hospital in May 2018. The intervention was conceived as a non-restrictive intervention including a multidisciplinary antimicrobial stewardship (AS) team (including Infectious Diseases, Microbiology, Pharmacy, Infection Prevention and Control, Hospital Epidemiology, and Psychology specialists) with the main aim of reducing overall antimicrobial consumption and MDROs infections in our unit. Quality Improvement certified guidelines for empirical antibiotic therapy were jointly drafted by the AS team and the ward physicians. Prescriptive appropriateness was systematically evaluated in terms of the following parameters: indication, choice of the molecule, dosage, duration, and use of combinations. Empirical prescriptions were evaluated based on the local guidelines for empirical treatment, in vitro activity, and de-escalation for targeted treatment. Infection control measures as rectal swabs for MDROs in all admitted patients, with isolation of positive patients, hand washing education, culture collection before starting antibiotics were also adopted [23].

The most frequently prescribed antibiotics were beta-lactam with beta-lactamase inhibitors. Potential toxicity was evaluated on a case-by-case basis and agents with a well-known hepato-toxicity (such as tigecycline) were rarely prescribed.

## Statistical analysis

Continuous variables are presented as mean ± standard deviation (SD) or median with interquartile range (IQR) based on data distribution. Categorical variables are expressed as frequency numbers and percentages. Either one-way ANOVA or Kruskal Wallis one-way ANOVA were used to compare continuous variables according to the data distribution pattern (normal or not normal). Categorical variables were compared using the Chi-square test. Both logistic and linear multivariate regression analyses were performed to determine if the following anamnestic or clinical variables: age, sex, BMI, presence of infection, MDR infection, white blood cells count, could be independently associated with liver complications.

The variable selection was done through sequential replacement (a stepwise method) which consists of a combination of backward and forward techniques. If the p-value was less than 0.05 or above 0.1 the covariates were respectively included and excluded from the regression model. No fixed variables were considered [24]. Statistical Package for Social Science (SPSS) version 22 was used for all data analysis. All tests were 2-sided, and p-values <0.05 were considered statistically significant.

## Results

229 patients were enrolled, 68.1% were males with a mean age of 64.4 ± 11.2 years. Table 1 describes the general characteristics of the population.

The cirrhosis pathogenesis was more virus related (HBV/HCV), follow by alcohol and NASH aetiology. In particular HBV was detected in 5.2% of whom 60% was treated with entecavir, while HCV was documented in 31% of whom all patients underwent direct acting antiviral agents therapy with sustained virological response.

During the follow-up, we recorded 101 cumulative infectious events in 72 patients (31.5% of the studied population). 32 out of 101 infections (31.7%) occurred in patients who had at least one previous infectious episode. In 48% of cases, infections were found to be community acquired, 25% were nosocomial and 27% were healthcare-associated infections. Regarding nosocomial infections, 6.6% of the patients were carriers of healthcare devices, such as urinary catheters, central venous catheters (CVC), or other drainage tubes. No differences in the number of infections were documented between the years of follow up (about 9.5% of infections per single year).

Sepsis was the most frequent infection (24.7%), followed by pneumonia (19.8%), and SBP (17.8%). UTIs were the 11.9%, while other recorded infections were cholangitis (3%), gastroenteritis (2%) and SSTI (3%) (Table 1.2 in S1 File). We also identified 15 episodes of bacterascites and 25 episodes of asymptomatic bacteriuria (not considered in the further analysis). In 41 cases out of 101 total infective episodes (40.6%), there was a microbiological isolation at blood cultures. The microorganism most frequently isolated in blood cultures was *S. aureus*, followed by *E. faecalis*, *S. haemolyticus*, *S. epidermidis*, *P. aeruginosa* and *Salmonella spp*.

*Staphylococcus spp.*, *E. coli* and *Enterococcus spp.* were the organisms most frequently isolated by ascitic cultures while *E. coli*, *P. mirabilis* and *Enterococcus spp.* were the organisms most frequently isolated by urine.

**Table 1. Baseline characteristics of the population at the enrolment (N: 229 patients).**

| | |
|---|---|
| Age (years), mean ± SD | 64.4 ± 11.2 |
| Male sex, n (%) | 156 (68.1) |
| **Comorbidities** | |
| Diabetes, n (%) | 71 (31.0) |
| CKD, n (%) | 21 (9.2) |
| Hypertension, n (%) | 101 (44.1) |
| Hyperlipidemia | 28 (12.2) |
| COPD, n (%) | 8 (3.5) |
| **Cirrhosis aetiology** | |
| Alcohol, n (%) | 53 (23.0) |
| HCV-related, n (%) | 71 (31.0) |
| HBV-related, n (%) | 12 (5.2) |
| NASH, n (%) | 18 (7.8) |
| Viral hepatitis + alcohol, n (%) | 26 (11.4) |
| Other, n (%) | 49 (21.6) |
| **Liver Functional Scores** | |
| MELD score, mean ± SD | 10.0 ± 4.2 |
| Child-Pugh score B and C, % | 17.7 |

CKD, chronic kidney disease; COPD, chronic obstructive pulmonary disease; HCV, hepatitis C virus; NASH, non-alcoholic steatohepatitis; HBV, hepatitis B virus; MELD, model for end-stage liver disease

**Table 2. Baseline characteristics in not infected and infected patients.**

| Total cohort (n = 229) | Infected (n = 72) | Not infected (n = 157) | p-value |
|---|---|---|---|
| Age (years), mean ± SD | 66.2 ± 12.9 | 63.6 ± 10.2 | 0.100 |
| Male sex, n (%) | 46 (63.9) | 110 (70.1) | 0.350 |
| **Comorbidities** | | | |
| Diabetes, n (%) | 28 (38.9) | 43 (27.4) | 0.080 |
| CKD, n (%) | 9 (12.5) | 12 (7.6) | 0.230 |
| Hypertension, n (%) | 34 (47.2) | 67 (42.7) | 0.520 |
| Hyperlipidemia, n(%) | 14 (19.4) | 14 (8.9) | **0.020** |
| COPD, n (%) | 2 (2.8) | 6 (3.8) | 0.690 |
| Antibiotic prophylaxis, n (%) | 27 (27.0) | 29 (18.4) | **0.004** |
| **Liver Functional scores** | | | |
| MELD score, mean ± SD | 11.8 ± 5.9 | 9.3 ± 2.9 | **0.002** |
| Child-Pugh score B and C, (%) | 36 | 9.7 | **< .001** |
| **Blood tests** | | | |
| WBC (x 10^9/L), mean ± SD | 5.96 ± 2.5 | 5.13 ± 2.3 | 0.080 |
| Neutrophiles (x 10^9/L), mean ± SD | 3.59 ± 2.0 | 3.12 ± 1.7 | 0.200 |
| Lymphocytes (x 10^9/L), mean ± SD | 1.35 ± 0.5 | 1.39 ± 0.6 | 0.810 |
| Platelets (x 10^9/L), mean ± SD | 147 ± 112 | 116 ± 55 | **0.030** |
| Hemoglobin (g/dL), mean ±SD | 12.1 ± 2.2 | 13.3 ± 2.1 | **0.006** |
| CRP (mg/dL) | 45±10.0 | 2.5±0.6 | **0.001** |
| **Complications** | | | |
| ACLF, n (%) | 11 (15.3) | 15 (9.6) | 0.200 |
| PVT, n (%) | 17 (23.6) | 22 (14.0) | **0.050** |
| Bleeding, n (%) | 14 (19.4) | 18 (11.5) | 0.100 |
| HRS, n (%) | 11 (15.3) | 5 (3.2) | **< .001** |
| Ascites, n (%) | 53 (73.6) | 60 (38.2) | **< .001** |
| HE, n (%) | 22 (30.6) | 19 (12.1) | **< .001** |
| Portal hypertension, n (%) | 51 (70.8) | 116 (73.9) | 0.630 |
| Outcome death, n (%) | 35 (48) | 28 (17) | **0.004** |
| **MDROs** | | | |
| MDROs on surveillance enteric swabs, n (%) | 8 (11.1) | 8 (5.1) | **0.030** |

CKD, chronic kidney disease; COPD, chronic obstructive pulmonary disease; MELD, model for end-stage liver disease; WBC, white blood cells; CRP, c-reactive protein; ACLF, acute-on-chronic liver failure; PVT, portal vein thrombosis; HRS, hepatorenal syndrome; HE, hepatic encephalopathy.

No fungi and anaerobic microorganisms were detected.

Differences at enrolment were: antibiotic prophylaxis (27% vs 18.4%, p = 0.004), MELD score (11.8 ± 5.9 vs 9.3 ± 2.9, p = .002) and Child-Pugh (CP) B—C (36% vs 9.7%, p < .001), that were significantly higher in infected patients compared to the uninfected ones. Similarly, platelets count was significantly higher (147x$10^3$ ± 112 vs 116x$10^3$ ± 55, p = .003), while haemoglobin levels were significantly lower (12.1 ± 2.2 vs 13.3 ± 2.1, p = .006) in patients with infections compared to those uninfected (Table 2).

## MDROs infections

15 patients (20.8%) had at least one infection sustained by MDROs, with an incidence rate for year of 3.5%. The most frequently detected MDROs in bloodstream were methicillin-resistant (MR) coagulase-negative Staphylococci, followed by MDR *Pseudomonas spp*. and VRE, while in urine were MDR *Proteus*, MDR *Pseudomonas spp*., MDR *K. pneumonie* and VRE.

**Table 3. Comparison of baseline characteristics of infected patients with MDROs and infected patients without MDROs.**

|  | MDROs infections (n = 15) | Not MDROs infections (n = 57) | p-value |
|---|---|---|---|
| Age, mean ±SD | 65.9 ±12.7 | 66.3 ±13.2 | 0.748 |
| Male sex, n (%) | 13 (86.7) | 33 (57.9) | 0.06 |
| **Comorbidities** |  |  |  |
| Diabetes, n (%) | 4 (26.7) | 24 (42.1) | 0.282 |
| COPD, n (%) | 2 (13.3) | 0 (0) | **0.005** |
| CKD, n (%) | 2 (13.3) | 7 (12.3) | 0.914 |
| Hypertension, n (%) | 6 (40) | 28 (49.1) | 0.536 |
| Antibiotic prophylaxis n(%) | 5 (33) | 9 (15) | **0.005** |
| **Blood tests** |  |  |  |
| WBC (x 10^9/L), median (IQR) | 6.34 (3.4) | 5.82 (2.2) | 0.595 |
| Neutrophiles (x 10^9/L), median (IQR) | 4.20 (23.6) | 3.44 (1.4) | 0.397 |
| Lymphocytes (x 10^9/L), median (IQR) | 1.49 (0.5) | 1.34 (0.6) | 0.644 |
| Platelets (x 10^9/L), mean ±SD | 111 ±58 | 161 ±125 | **0.023** |
| Rectal MDROs swab (n) | 3 (20.0) | 6 (10.5) | 0.167 |
| **Liver Functional scores** |  |  |  |
| MELD score, mean ± SD | 12.6 ± 6.1 | 8.7 ± 2.4 | **0.05** |
| Child-Pugh score B and C, (%) | 41 | 12 | **< .001** |
| **Complications** |  |  |  |
| ACLF, n (%) | 4 (26.7) | 7 (12.3) | 0.176 |
| PVT, n (%) | 3 (20.0) | 14 (24.6) | 0.716 |
| bleeding, n (%) | 3 (20.0) | 11 (19.3) | 0.952 |
| HRS, n (%) | 4 (26.7) | 7 12.3) | 0.173 |
| Outcome death, n (%) | 6 (40.0) | 28 (49.1) | 0.06 |

COPD, chronic obstructive pulmonary disease; CKD, chronic kidney disease; WBC, white blood cells; MELD, model for end-stage liver disease; ACLF, acute-on-chronic liver failure; PVT, portal vein thrombosis; HRS, hepatorenal syndrome.

Regarding ascitic fluid MDROs, an extended spectrum beta-lactamase (ESBL)-producing *E. coli* and MR coagulase-negative Staphylococci were isolated, while no CRE, VRE and MR *Staphylococcus aureus (MRSA)* were isolated in our ascitic samples. Only one bronchoalveolar lavage was conducted with a MDR *Pseudomonas* isolated. (Table 1.3 in S1 File) 42.1% of these infections were sepsis, followed by SBP (26.3%), pneumonia (15.8%) and UTIs (15.8%). 16 patients among the whole population (6.9%) were found to be MDROs carriers at rectal swabs.

No significant differences were detected comparing MDROs-infected patients with non-MDROs infected ones, except for COPD, platelets value and antibiotic prophylaxis use. No differences were observed regarding MELD and CP score (Table 3).

## MDROs and SAVE

A progressive decrease in the occurrence of infections sustained by MDROs was observed during the follow-up period, concurrently with the introduction in our Institution of an AS program.

In particular the incidence rate of total infections increased from 16.6 infections/100 person/year in the two- years 2017–18 up to 18.7 infections/100 person/year in the three years 2019–2021 (p = ns), while the incidence rate of MDROs decreased from 4.2 infections/100 person/year in the two-years 2017–18 to 1.4 infections/100 person/year in the three years 2019–2021 (IRD 28.6; 95% CI: 4.6–52.5, p = 0.02) (Fig 1).

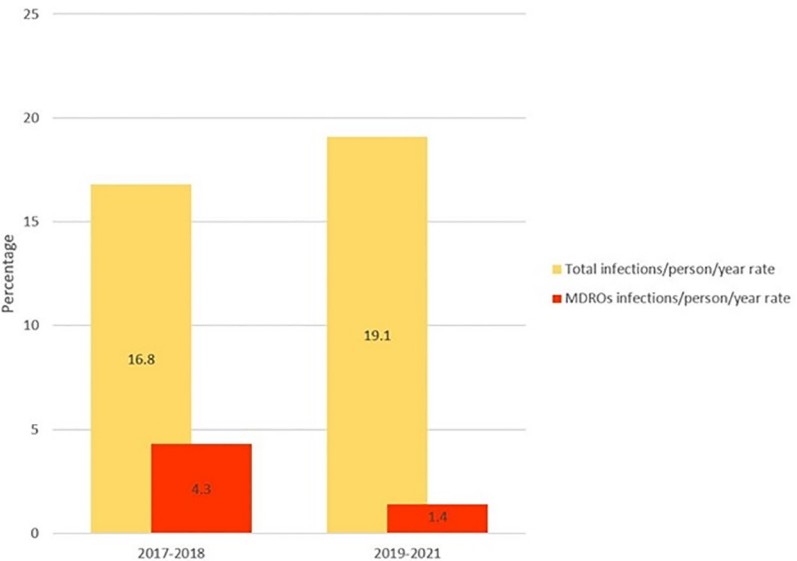

**Fig 1. Incidence rates for total infections (MDSOs and MDROs)/person/year and for MDROs infections/person/year.**

## Liver complications and mortality

Hepatic complications, such as ascites (73.6% vs 38.2%, p < .001), hepatorenal syndrome (HRS) (15.3% vs 3.2%, p < .001) and hepatic encephalopathy (HE) (30.6% vs 12.1%, p < .001) occurred significantly more frequently in infected than in non-infected patients, while portal vein thrombosis (PVT) nearly reached significance (23.6% vs 14%, p = .05). Bleeding from gastric or oesophageal varices, acute-on-chronic liver failure (ACLF) and portal hypertension were not significantly different in the two subgroups (Table 2). Ascites was significantly associated with a previous infection on multivariate analysis (Table 4). Regarding MDROs infections, no differences in liver complications or mortality were documented.

**Table 4. Multivariate analysis with infections (both MDROs and multi-drugs sensible organisms) as dependent variable.**

| Variable | Multivariate | |
| --- | --- | --- |
| | OR (95% CI) | p-value |
| Age | 1.01 (0.97–1.06) | 0.544 |
| Male sex | 0.91 (0.33–2.47) | 0.858 |
| Diabetes | 2.05 (0.77–5.43) | 0.149 |
| CKD | 0.68 (0.12–3.83) | 0.662 |
| MELD score | 1.11 (0.96–1.28) | 0.149 |
| CP score | 1.58 (0.46–5.40) | 0.462 |
| History of ascites | 4.27 (0.77–2.14) | **<0.001** |
| PH during the 5 years | 0.61 (0.19–1.97) | 0.416 |
| HCC at baseline | 0.87 (-1.00–0.73) | 0.757 |
| Antibiotic prophylaxis during the 5 years | 0.55 (0.14–2.12) | 0.382 |

CKD, chronic kidney disease; MELD, model for end-stage liver disease; CP, Child Pugh; PH, portal hypertension; HCC, hepatic cell carcinoma

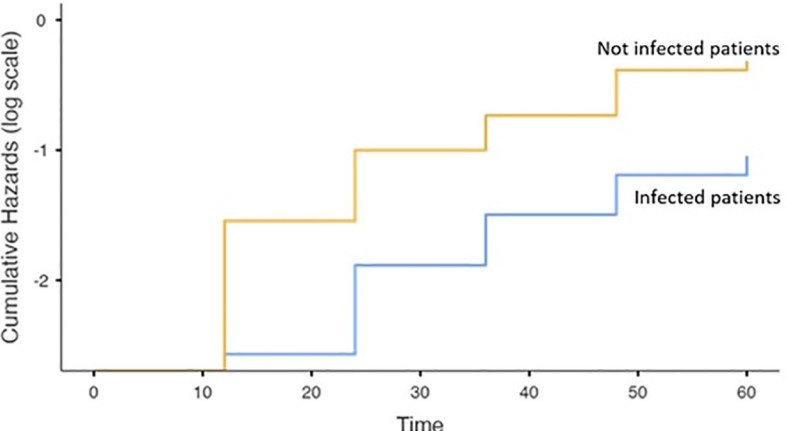

**Fig 2. Survival cumulative Hazards between patients with and without infections.** Yellow line: not infected patients, blue line: infected patients.

68 patients (29.7%) died during the observation period and 13 patients (6%) underwent liver transplantation. Risk factors for death were age, diabetes and bacterial infections (OR 3.30, CI 95%: (1.63–6.70), with higher risk in case of infections sustained by MDROs (OR 4.45, CI 95%: 1.131–10.528; Table 4.1 in S1 File).

The survival rate of infected patients was significantly lower compared to not infected one as shown in Fig 2, while the survival rate was similar between the MDROs infection and non-MDROs infection subgroups (Fig 3 and Table 4.1 in S1 File).

## Discussion

Bacterial infections are a major cause of morbidity and mortality in cirrhotic patients, in particular if sustained by MDROs [19].

As suggested by the current guidelines [8] a strict analysis of the microbiological background on a regional and even local basis is the main goal to better manage therapy through antibiotic stewardship (correct choice, administration timing, de-escalation). The main aim of our study was to evaluate the incidence of infections in cirrhotic patients belonging to our

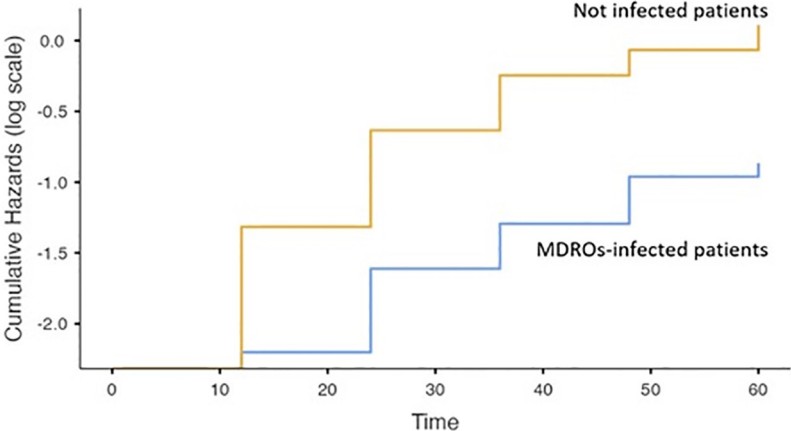

**Fig 3. Survival cumulative Hazards between patients with MDROs infections and not infected.** Yellow line: not infected patients, blue line: MDROs-infected patients.

centre, in particular caused by MDROs, and verify the efficacy of local AS introduced in 2018. In addition, we wanted to confirm the potential associations between bacterial infections, acute liver complications and crude mortality.

31% of our patients developed a bacterial infection, in line with literature (25–35%) [21], and 30% of bacterial infections occurred in patients with at least one previous infectious episode, confirming the trend towards infectious complications to recur in cirrhotic patients [17, 19].

The most common type of bacterial infection was sepsis (25%), followed by pneumonia (20%) and SBP (18%). This result differs from what has been reported in some previous studies, especially regarding the relative frequencies of sepsis (25% vs 8–15%) and SBP (18% vs 20–30%). However, in our population the higher incidence of sepsis may be related to the high number of blood cultures performed to detect bacterial infections as requested by our antimicrobial stewardship program.

Furthermore, in contrast with the study of Lingiah and al. [25], showing a higher incidence of infections, we excluded asymptomatic bacteriuria and bacterascites from the main analysis, as our SAVE guidelines do not recommend antibiotic therapy for them, in line with other similar guidelines (except for clinical features suggestive for infection) [26].

Compared to other studies, our data showed a lower incidence of MDROs infections during the whole follow up period and in particular after the introduction of the AS (the incidence rate of MDROs decreased from 4.2% in the biennium 2017–18 to 1.4% in the three-year period 2019–21). Piano et al. [17] in a recent multicentre study showed that the overall incidence of multidrug-resistant infections in cirrhotic patients was 34% and that the incidence varied significantly comparing different geographical areas, with the highest prevalence of MDROs in India (73%). In another recent study by Fernandez et al. [16], the overall prevalence of MDR infections was 29% in 2011, rising to 38% in 2018 [18].

In our study the lower MDROs infections incidence of 20.8%, probably could be considered as a result of the antibiotic stewardship started in our canter and a sign of avoiding the antibiotic overuse, de-escalation therapy and the strict management on ward of colonized patients to keep MDROs as limited as possible.

Our data are the result of the strict contact isolation procedures of colonized/infected patients adopted on our ward to keep MDROs as limited as possible. The AS has allowed us to review the local microbial epidemiology, in particular comparing other studies conducted in Italy, where the approach to antibiotic therapy should take into account a high incidence of MDROs [3]. In our local setting instead, this choice could lead to an increase in the local antibiotic resistance and MDROs incidence itself. To confirm that, we had also documented a higher incidence of MDROs in patients with previous use of antibiotic prophylaxis.

In our study, MR coagulase-negative Staphylococci were the most frequently isolated MDROs (30%), while gram-negatives, including *E. coli* producing ESBL, MDR *Pseudomonas spp.*, *K. pneumoniae* and *Proteus mirabilis*, accounted for 43% of the overall isolated microorganisms. These data are consistent with the epidemiological change observed in recent years, which has led to an increase in the frequency of gram-positive infections, probably also due to the use of norfloxacin as prophylaxis for SBP in cirrhotic patients at high risk of infection [27]. In contrast, *E. coli* producing ESBL was rarely isolated (8.5% of MDR infections) [16, 17, 28, 29].

The average annual mortality rate in patients with bacterial infections was found to be higher than in patients without infection, (OR 3.30, CI 95%: (1.63–6.70), especially for MDROs infections (OR 4.45, CI 95%: 1.131–10.528; see Table 4), in agreement with what has been reported in the literature [12].

The novelty of our study is a decrease in MDROs incidence between years 2017–2018 and 2019–2021, which reflects a first impact on the local epidemiology of the SAVE project introduced in the Verona hospital in 2018.

Risk factors for the development of infections were found to be the severity of the liver disease and previous infections. Finally, it was confirmed that bacterial infections, in particular those caused by MDROs bacteria, are an important cause of adverse prognosis, suggesting the addition of previous infectious data to prognostic models such as the CP or MELD score because it could improve clinical application.

Our study presents some limits: first of all, the analysis resulted in the lack of some patients' data we could not retrieve for our analysis, in particular the microbiological cultures were not always systematically collected, especially blood cultures before starting the empiric antibiotic treatment. Despite the differences in MDROs in patients with prophylaxis showed in our study, we are not able at the moment to confirm the possible correlation with MDROs, due to the low number of patients.

Secondly, we are not able yet to demonstrate the effect on mortality of our antimicrobial stewardship program for the short follow up from its introduction (2018).

The strength of our study is the long follow-up and the effort made to identify and analyse only clinically symptomatic infections, excluding colonization. In particular the study of Lingiah and al. [25] considered in the same group asymptomatic and symptomatic infections all together.

## Conclusions

In conclusion, our study confirms the high incidence of bacterial infections in the clinical history of cirrhotic patients, in particular in case of MDROs bacteria, and the high incidence of complications linked with infections. It's crucial the punctual definition of the local epidemiology in terms of most frequent microrganisms and relative antibiotic resistance to rapidly start the most appropriate empiric therapy and subsequently targeted treatment, in line with AS indications.

## Supporting information

**S1 File.**
(DOCX)

**S1 Checklist.**
(DOCX)

## Acknowledgments

We acknowledge the contribution of all the following members of the SAVE team: Federico Bercelli, Chiara Bovo, Maria Cristina Caldana, Lidia Dal Piccolo, Valeria Donisi, Silvia Manfrè, Paola Marini, Annarita Mazzariol, Mara Merighi, Giorgio Nicolis, Michela Rimondini, Alice Sartori, Patrizia Soffiatti, Fabio Soldani, Livia Tiro, Chiara Tonolli.

## Author Contributions

**Conceptualization:** Andrea Dalbeni, Elena Carrara, David Sacerdoti.

**Data curation:** Anna Mantovani, Donatella Ieluzzi, Evelina Tacconelli, David Sacerdoti.

**Formal analysis:** Mirko Zoncapè, Michele Bevilacqua.

**Funding acquisition:** Leonardo De Marco, Veronica Paon, David Sacerdoti.

**Investigation:** Anna Mantovani, Leonardo De Marco, Donatella Ieluzzi.

**Methodology:** Anna Mantovani, Filippo Cattazzo.

**Supervision:** Michele Bevilacqua, Anna Maria Azzini, David Sacerdoti.

**Validation:** Anna Maria Azzini, David Sacerdoti.

**Visualization:** Mirko Zoncapè.

**Writing – original draft:** Andrea Dalbeni, Anna Mantovani.

**Writing – review & editing:** Filippo Cattazzo, Michele Bevilacqua, Veronica Paon, Anna Maria Azzini, Elena Carrara, Evelina Tacconelli, David Sacerdoti.

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
