## [Decision Letter · Decision Letter 0]

4 Nov 2022

PONE-D-22-25441The multi-drug resistant organisms infections decrease during the antimicrobial stewardship era in cirrhotic patients: an Italian cohort studyPLOS ONE

Dear Dr. Mantovani,

Thank you for submitting your manuscript to PLOS ONE. After careful consideration, we feel that it has merit but does not fully meet PLOS ONE’s publication criteria as it currently stands. Therefore, we invite you to submit a revised version of the manuscript that addresses the points raised during the review process.

Please address the questions and comments provided by the reviewers and revise the manuscript accordingly. 

We look forward to receiving your revised manuscript.

Kind regards,

Iddya Karunasagar

Academic Editor

PLOS ONE

Journal Requirements:

 "no.The funders had no role in study design, data collection and analysis, decision to publish, or preparation of the manuscript." 

5. Please include a copy of Tables 1.2 and 4.1 which you refer to in your text on pages 10 and 12, respecitvely.

Additional Editor Comments:

Please see the reviewer comments. Please address these, provide clarifications and revise the manuscript accordingly.

Reviewers' comments:

Reviewer's Responses to Questions

**Comments to the Author**

1. Is the manuscript technically sound, and do the data support the conclusions?

Reviewer #1: Yes

Reviewer #2: Yes

2. Has the statistical analysis been performed appropriately and rigorously? 

Reviewer #1: Yes

Reviewer #2: Yes

3. Have the authors made all data underlying the findings in their manuscript fully available?

Reviewer #1: Yes

Reviewer #2: Yes

4. Is the manuscript presented in an intelligible fashion and written in standard English?

Reviewer #1: No

Reviewer #2: Yes

5. Review Comments to the Author

Reviewer #1: The title of the article to be made meaningful to reflect the objectives of the study because its an association between MDRO infection and antibiotic stewardship program in th hospital.

The manuscript needs spell check eg: Staphilococci, cube mm to be written in standard format.

In results its not clear all the patient treatment were according to the antibiotic policy of the hospital and how is it captured.

Other compounding factors like infection control practices were followed or not because decrease in MDRO infection is multi factorial which cannot be attributed to a single program

Reviewer #2: The manuscript details a study of MDRO's identified in a cohort of patients with liver disease and the outcome measures of antimicrobial stewardship on these patients. While the study has enrolled 229 patients over a 5 year period, it is a single center data and hence may have bias.

The manuscript needs to be rephrased in certain sections to improve understanding. However, the following need to be addressed for clarity.

Introduction says that all patients were followed up for 5 years. But this study though started in 2017 also recruited patients in 2019 and 2020 who do not have a 5 year follow-up. Hence this needs to be clarified.

Organism nomenclature needs to be looked into. Enterobacteriaceae are now called Enterobacterales. E.coli and Staphylococci spellings are wrong.

Since this was a single center study, did this conform with National guidelines from Italy?

The difference between nosocomial infections and healthcare associated infections is explained by the authors. But do they conform with international definitions.? Infections in the study are classified as community acquired, nosocomial and hospital acquired. Clarity needs to be provided for the same with a reference for the classification

The authors have explained as Methicillin resistant Staphylococci as commonly isolated from urine cultures. Most of the population studied are males (68%) where Staphylococci in urine are not so common. The authors need to detail on the organisms isolated from urine.

Since pneumonias were the second most common infection isolated, the authors need to explain what the profile of organisms isolated from these specimens. A table of organisms from lower respiratory specimens like BAL or ET secretions may add value to the paper.

Since Hepatitis B and C were the most common predispositions to the liver disease, it would be useful to add information on the number of patients on antiviral therapy and the choice of antivirals. Was genotypic analysis and serial viral loads done for these patients?

Though gut carriage of MDRO was taken as an infective event, what were the infection control measures instituted in these patients? Were these patients cohorted or isolated during the hospital stay?

Was any diagnostic stewardship measure included by the stewardship team? What were the correlation with inflammatory markers like PCT or CRP

What were the most common antibiotics used in the cohort studied and if any hepatic toxic antibiotic was used for treatment that impacted outcomes of the patients.

6. PLOS authors have the option to publish the peer review history of their article (what does this mean?). If published, this will include your full peer review and any attached files.

Reviewer #1: No

Reviewer #2: **Yes: **Anusha Rohit

---

## [Author Response · Author response to Decision Letter 0]

16 Dec 2022

Rebuttal Letter

We addressed each point raised by the academic editor in the revised manuscript. 

We thank the reviewer for the thoughtful inputs to our manuscript. Below, you will find a point-by-point reply to all the raised points, which have been addressed and highlighted in the revised manuscript with tracked changes. In addition, we have corrected some typing errors in the manuscript.

Reviewer #1: The title of the article to be made meaningful to reflect the objectives of the study because its an association between MDRO infection and antibiotic stewardship program in the hospital.

The manuscript needs spell check eg: Staphilococci, cube mm to be written in standard format.

In results its not clear all the patient treatment were according to the antibiotic policy of the hospital and how is it captured.

Other compounding factors like infection control practices were followed or not because decrease in MDRO infection is multi factorial which cannot be attributed to a single program

Thanks, we really appreciate this comment. We further specified in the text that our antibiotic stewardship consists not only of local antibiotic prescribing guidelines, but also of infection control measures such as performing rectal swabs for MDROs in all inpatients, with isolation of positive patients, handwashing education, collection of cultures before starting antibiotics, antibiotic initiation/change, dosing and duration, introduction of combinations, de-escalation, “switch-to-oral” strategies. We have added a reference in the text to better explain how the stewardship was organized and the result obtained. We agree that the decrease in MDRO infection is multifactorial and, from our point of view, due to the overall program. A spell check was made to correct organism nomenclature and the some typing errors. 

Carrara E., Sibani M., et all, “How to 'SAVE' antibiotics: effectiveness and sustainability of a new model of antibiotic stewardship intervention in the internal medicine area”, Int J Antimicrob Agents. 2022 Sep 11;106672. doi: 10.1016/j.ijantimicag.2022.106672.

Reviewer #2: The manuscript details a study of MDRO's identified in a cohort of patients with liver disease and the outcome measures of antimicrobial stewardship on these patients. While the study has enrolled 229 patients over a 5 year period, it is a single center data and hence may have bias.

The manuscript needs to be rephrased in certain sections to improve understanding. However, the following need to be addressed for clarity.

Introduction says that all patients were followed up for 5 years. But this study though started in 2017 also recruited patients in 2019 and 2020 who do not have a 5 year follow-up. Hence this needs to be clarified.

We thank the reviewer for this comment, we better clarified in the text that the study period was of 5 years, with a mean follow up time of 42.7 months.

Organism nomenclature needs to be looked into. Enterobacteriaceae are now called Enterobacterales. E.coli and Staphylococci spellings are wrong.

Thanks, we have corrected the organism nomenclature.

Since this was a single center study, did this conform with National guidelines from Italy?

We thank the reviewer for the question, unfortunately Italy does not have national guidelines for empirical antibiotic treatment in hospitalized patients. The guidelines promoted during the project were drafted in accordance with international guidelines tailored to local antimicrobial resistance data. 

The difference between nosocomial infections and healthcare associated infections is explained by the authors. But do they conform with international definitions? Infections in the study are classified as community acquired, nosocomial and hospital acquired. Clarity needs to be provided for the same with a reference for the classification

The definition used reflected standard international definition as in ‘Collins, A. S., & Hughes, R. (2008). Patient safety and quality: an evidence-based handbook for nurses. Preventing health care–associated infections.’ The reference has been added in the paper.

The authors have explained as Methicillin resistant Staphylococci as commonly isolated from urine cultures. Most of the population studied are males (68%) where Staphylococci in urine are not so common. The authors need to detail on the organisms isolated from urine.

Thanks for the suggestion; we had clarified in the “MDROs infection” paragraph the most frequently isolated MDROs and added table 1.3.

Since pneumonias were the second most common infection isolated, the authors need to explain what the profile of organisms isolated from these specimens. A table of organisms from lower respiratory specimens like BAL or ET secretions may add value to the paper.

Thanks for the suggestion, to better explain this concept we added Table 1.3 to the text.

Since Hepatitis B and C were the most common predispositions to the liver disease, it would be useful to add information on the number of patients on antiviral therapy and the choice of antivirals. Was genotypic analysis and serial viral loads done for these patients?

We thank the reviewer for this comment. In Table 1 we reported the general characteristic of the population in particular cirrhosis aetiologies comprehensive of HBV and HCV infections. We added a sentence in the text to clarify that all patients from 2017 underwent direct acting antiviral agents therapy for eradication of HCV, while 60% of HBV were treated with entecavir (the other 40% had not indication for specific therapy).

Though gut carriage of MDRO was taken as an infective event, what were the infection control measures instituted in these patients? Were these patients cohorted or isolated during the hospital stay?

Our antimicrobial stewardship consists also in infection control measures: all patients resulted positive to a rectal swab for MDROs were kept isolated or cohorted. We added this concept in the text. 

Was any diagnostic stewardship measure included by the stewardship team? What were the correlation with inflammatory markers like PCT or CRP

The stewardship intervention targeted mostly appropriateness of antibiotic prescriptions and no specific diagnostic stewardship intervention was carried out. However, guidelines on the management of commonly encountered infections included some generic recommendations on appropriateness of cultures and diagnostic tests (beta-D-glucan, PCT, CRP..). We added in table 2 correlation with CRP.

What were the most common antibiotics used in the cohort studied and if any hepatic toxic antibiotic was used for treatment that impacted outcomes of the patients.

The antibiotics that were most prescribed were beta-lactam with beta-lactamase inhibitors. Potential toxicity was evaluated on a case-by-case basis and agents with a well-known hepatotoxicity (such as tigecycline) were rarely prescribed.

---

## [Decision Letter · Decision Letter 1]

2 Feb 2023

The multi-drug resistant organisms infections decrease during the antimicrobial stewardship era in cirrhotic patients: an Italian cohort study

PONE-D-22-25441R1

Dear Dr. Mantovani,

We’re pleased to inform you that your manuscript has been judged scientifically suitable for publication and will be formally accepted for publication once it meets all outstanding technical requirements.

Kind regards,

Iddya Karunasagar

Academic Editor

PLOS ONE

Additional Editor Comments (optional):

All reviewer comments have been addressed

Reviewers' comments:

Reviewer's Responses to Questions

**Comments to the Author**

1. If the authors have adequately addressed your comments raised in a previous round of review and you feel that this manuscript is now acceptable for publication, you may indicate that here to bypass the “Comments to the Author” section, enter your conflict of interest statement in the “Confidential to Editor” section, and submit your "Accept" recommendation.

Reviewer #1: All comments have been addressed

Reviewer #2: All comments have been addressed

2. Is the manuscript technically sound, and do the data support the conclusions?

Reviewer #1: Yes

Reviewer #2: Yes

3. Has the statistical analysis been performed appropriately and rigorously? 

Reviewer #1: Yes

Reviewer #2: Yes

4. Have the authors made all data underlying the findings in their manuscript fully available?

Reviewer #1: Yes

Reviewer #2: Yes

5. Is the manuscript presented in an intelligible fashion and written in standard English?

Reviewer #1: Yes

Reviewer #2: Yes

6. Review Comments to the Author

Reviewer #1: All the comments are addressed except the modification of title of the study. Title can be modified as reflecting the relationship between the MDRO infection and Antibiotic stewardship.

Reviewer #2: The authors have addressed all comments adequately and the paper can be accepted in its present form.

7. PLOS authors have the option to publish the peer review history of their article (what does this mean?). If published, this will include your full peer review and any attached files.

Reviewer #1: No

Reviewer #2: **Yes: **Anusha Rohit

---

## [Editor Report · Acceptance letter]

7 Feb 2023

PONE-D-22-25441R1 

The multi-drug resistant organisms infections decrease during the antimicrobial stewardship era in cirrhotic patients: an Italian cohort study 

Dear Dr. Mantovani:

I'm pleased to inform you that your manuscript has been deemed suitable for publication in PLOS ONE. Congratulations! Your manuscript is now with our production department. 

Kind regards, 

on behalf of

Dr. Iddya Karunasagar 

Academic Editor

PLOS ONE